# Ceria-Catalysed Production of Dimethyl Carbonate from Methanol and CO_2_: Effect of Using a Dehydrating Agent Combined with a Solid Cocatalyst

**DOI:** 10.3390/molecules29235663

**Published:** 2024-11-29

**Authors:** Dichao Shi, Svetlana Heyte, Mickaël Capron, Sébastien Paul

**Affiliations:** 1School of Chemical and Environmental Engineering, Liaoning University of Technology, Jinzhou 121001, China; shidichao@lnut.edu.cn; 2Univ. Lille, CNRS, Centrale Lille, Univ. Artois, UMR 8181—UCCS—Unité de Catalyse et Chimie du Solide, F-59000 Lille, France; svetlana.heyte@univ-lille.fr (S.H.); mickael.capron@univ-lille.fr (M.C.)

**Keywords:** dimethyl carbonate, CO_2_, methanol, ceria, TMM, 13X, cocatalyst

## Abstract

The direct synthesis of dimethyl carbonate (DMC) from CO_2_ and methanol over ceria-based catalysts, in the presence of a dehydrating agent shifting the thermodynamical equilibrium of the reaction, has received significant interest recently. In this work, several dehydrating agents, such as molecular sieves, 2,2-dimethoxypropane (DMP), dimethoxymethane (DMM) and 1,1,1-trimethoxymethane (TMM), are combined with commercial ceria to compare their influence on the DMC yield obtained under the same set of operating conditions. TMM is found to be the most efficient; however, its conversion is not complete even after 48 h of reaction. Therefore, it is proposed for the very first time, to the best of our knowledge, to add a second solid cocatalyst in the reaction medium to accelerate the TMM hydration reaction without degrading the DMC already formed. Basic oxides and acidic zeolites with different Si/Al ratios are employed to accelerate the hydration of TMM, so as to improve the DMC yield. 13X was identified as the best option to play this role. Finally, three different commercial cerias are tested in the presence of TMM and molecular sieve 13X as the second catalyst. The most efficient combination of ceria, TMM, and molecular sieve 13X is ultimately tested in a 250 mL autoclave to start to scale up the process. A very high DMC production of 199.5 mmol DMC/g_cat._ is obtained.

## 1. Introduction

Dimethyl carbonate (DMC), which contains methyl and carbonyl groups, is the simplest organic carbonate. Much attention has been paid to DMC because of its versatile chemical properties and various industrial applications [1,2,3,4,5,6]. Direct synthesis of DMC from CO_2_ and methanol is highly favourable among all the synthesis routes because of its compliance with the concept of green chemistry in terms of the absence of toxicity, low waste production, atom economy and abundance of the feedstock [7,8,9,10]. It also would be an elegant way to valorize CO_2_ in the frame of the so-called Carbon Capture and Utilization (CCU) concept [11,12]. Moreover, the low cost of the feedstock is in favour of such a development at the industrial scale.

CeO_2_ is the most efficient heterogeneous catalyst reported so far for the direct synthesis of DMC from CO_2_ and methanol due to its surface acid-base sites, which are important for the adsorption and activation of the reactants. Different morphologies of CeO_2_ possess different amounts of acid-base sites, leading to a variety of catalytic performances. It turned out that the spindle-like structure of CeO_2_ exhibited the highest yield to DMC because of the highest acidity and basicity of its sites compared to nano-octahedrons, nanorods, and nanocubes structures [13,14,15,16]. The combination of a second metal oxide with CeO_2_ to modify the acid-base properties, such as Al_2_O_3_, SiO_2_, molecular sieve 4A (further abbreviated as 4A) [17], ZnO [18], and CaO [19], is another way of improving DMC yield. However, even if many efforts were made, the results are at this date far from being satisfying. One of the main reasons is the thermodynamical limitation of the equilibrated reaction. This can be tackled by adding in the reaction medium a suitable dehydrating agent, which can adsorb or react with water to shift the equilibrium of the DMC formation reaction to the right side and hence improve the DMC production [8].

Molecular sieve 3A was used as an inorganic absorbent dehydrating agent in combination with a CeO_2_ catalyst due to the advantages of easy operation, no by-product formation and easy recycling. This combination leads to a production of 2 mmol DMC/g_cat._ at a reaction temperature of 120 °C, a pressure of 150 bar and a reaction time of 4 h [20]. This performance is not sufficient to meet the requirements of the industry as the adsorption can only take place at the interface between the solid and the liquid reactive media. Therefore, organic compounds were also considered dehydrating agents as they are solubilized in the reactive medium so they can react with water molecules as soon as they are formed. Recently, 2-cyanopyridine was used to promote the DMC yield. The results showed that almost full conversion of methanol and a very high DMC production of 138 mmol/g_cat._ were obtained under 50 bar of pure CO_2_ at 120 °C after 12 h of reaction time [21]. The main by-product was 2-picolinamide formed through hydration of 2-cyanopyridine. Based on the high yield of DMC when using 2-cyanopyridine as a dehydrating agent, many studies have been conducted and all of them reported excellent DMC yields [14,22,23,24,25,26]. Nevertheless, the main drawback of using 2-cyanopyridine as a dehydrating agent is the hazardous character of this molecule and hence the difficulty of operation, which limits its application in industry. Alternative green dehydrating agents that are not toxic are more favourable and compliant with the green chemistry rules. Honda et al. found that CeO_2_ catalyst showed good performances in the presence of acetonitrile (24 mmol DMC/g_cat._ under 5 bar at 150 °C and after 24 h of reaction) and of benzonitrile (48.2 mmol DMC/g_cat._ under 5 bar at 150 °C and after 24 h of reaction) [27,28]. Tomishige et al. used 2,2-dimethoxypropane (DMP) as a dehydrating agent combined with a CeO_2_-ZrO_2_ catalyst. They reported 13.8 mmol DMC/g_cat._ under 60 bar at 110 °C and after 140 h of reaction [29]. 1,1,1-trimethoxymethane (TMM) also has been employed as a dehydrating agent on Ce–Zr oxide/graphene catalyst by Saada et al. The DMC production was 51.6 mmol DMC/g_cat._ under 275 bar at 110 °C and after 16 h of reaction time [30]. However, no firm conclusion has been drawn on which dehydrating agent combined with CeO_2_ catalyst is optimal due to the variation in reaction conditions, such as the amount of reactant, pressure, temperature, and reaction time.

A novel concept consisting of adding a second catalyst to promote the hydration reaction of the dehydrant agent was proposed by Choi et al. [31]. A DMC production of 40 mmol DMC/g_cat._ was obtained under 300 bar at 180 °C and after 24 h of reaction over Bu_2_SnO catalyst in the presence of DMP by adding a strong acid [Ph_2_NH_2_]OTf as the second catalyst. This was more than double the DMC production without adding a second catalyst (17 mmol DMC/g_cat._). It should be noted that this result was obtained under very harsh conditions. Nevertheless, this work showed an interest in adding a second catalyst in the reaction medium to accelerate the dehydrant agent reaction with water hence improving the DMC production. Obviously, a solid catalyst would be preferable regarding the easiness of its recovery from the reactive medium after the reaction.

Based on the reported literature, DMC production is far from the industrial demand in terms of efficiency. Therefore, the main objective and innovation of this study are to increase the production of DMC and explore ways to start the scale-up of the process. In this work, a series of CeO_2_ catalysts were tested in combination with various dehydrating agents under the same set of operating conditions to synthesize DMC directly from CO_2_ and methanol. Afterwards, with the most efficient dehydration agent, the effect of adding a second catalyst to accelerate the hydration of the selected dehydrant agent was also screened. Finally, the efficient combination of ceria, dehydrant agent, and a second catalyst started to scale up the process.

## 2. Results and Discussion

### 2.1. Effect of Reaction Temperature

The catalytic performance of the CeO_2_ catalyst supplied by Daiichi Kigenso Kagaku Kogyo Co. Ltd. company (Osaka, Japan) was examined for the direct synthesis of DMC from methanol and CO_2_. In order to identify the optimal reaction temperature for a high DMC production, different reaction temperatures have been used to evaluate the catalytic activity, as shown in Figure 1.

DMC production increased dramatically with increasing the reaction temperature, and it reached 1.1 mmol DMC/g_cat._ at 130 °C. Higher reaction temperatures are more favourable for the formation of dimethyl ether (DME) [32]. Moreover, thermodynamics shows that high reaction temperature is not in favour of the formation of DMC [33]. Therefore, the reaction temperature for the rest of the study was fixed at 130 °C. In terms of pressure, high CO_2_ pressure is more favourable for the formation of DMC as the positive Gibbs free-energy change decreases with pressure according to the thermodynamic equations [34]. Therefore, in this study, the pressure was initially set at 20 bar (the working pressure was 38 bar during the reaction due to the heating of the closed vessel). Therefore, the reaction conditions in this work are finally 130 °C, 38 bar CO_2_ pressure, and 30 mg of catalyst.

### 2.2. Selection of the Dehydrating Agent

As said above, the finding of an efficient dehydrating agent for the in situ water removal and reaction equilibrium shift to promote the DMC yield is necessary. Therefore, several liquid or solid dehydrating agents were screened in combination with CeO_2_, as shown in Figure 2.

The results indicate that Z1, Z2, Z3, 5A and 13X do not effectively promote DMC production. Conversely, they even induce the decomposition of DMC because of their excessive acidic sites, resulting in lower DMC productions than those obtained in the blank test without any dehydrating agent over the CeO_2_ catalyst.

Liquid dehydrating agents, such as TMM, DMP, and DMM, were also tested and proved to be more efficient. In all three cases, the methanol/liquid dehydrating agent was fixed to 6:1 based on previous studies, such as Tomishige et al. [29,35].

It turned out that TMM is the most efficient dehydrating agent to promote the DMC production when combined with CeO_2_ catalyst, the DMC production can reach up to 34 mmol/g_cat._. It demonstrates that TMM is a suitable dehydrating agent for the reaction. It is interesting to note that TMM is of particular interest as it releases methanol in the reaction medium when hydrolyzed. This supplementary methanol can, of course, be seen as a supplementary reactant for DMC formation.

One may suppose that lowering the methanol/TMM molar ratio, keeping the volume of methanol and increasing the amount of TMM, would be preferable as TMM can react with more water formed. Hence, the optimization of the molar ratio of methanol and TMM was further studied, as shown in Figure 3.

The DMC production obtained with a methanol/TMM molar ratio of 6:1 (34 mmol DMC/g_cat._) is close to that obtained at a molar ratio of 2:1 (26.8 mmol DMC/g_cat._). The TMM conversion, respectively, decreases from 61% to 12%. This shows that the TMM hydration rate is slow and that adding more TMM is not sufficient to enhance DMC production.

In order to study the kinetics of the reaction, the reaction time was extended from 6 to 48 h for the CeO_2_ catalyst with TMM as a dehydrating agent (molar ratio of methanol/TMM = 2:1) (Figure 4). A 15 mmol/g_cat._ increase in the DMC production is observed using CeO_2_ catalyst for a first 6 h-extension of the reaction time (12 h instead of 6 h initially). Then, DMC production reaches up to 73.3 mmol/g_cat._ after 48 h of reaction. There was also an increase in TMM conversion, which is an indication of more water consumption by TMM and a consequent increase in DMC formation. Nevertheless, a lot of TMM remains in the reaction medium even after 48 h (TMM conversion is then only 41%). Therefore, adding a second solid catalyst in the medium to promote the hydration of TMM is proposed to accelerate what is believed to be the limiting step of the whole process. The difficulty of selecting a second catalyst is here to avoid the degradation of DMC while accelerating the hydration of TMM.

### 2.3. Selection of a Second Catalyst

As discussed above, the production of DMC on CeO_2_ catalyst combined with TMM as a dehydration agent can reach up to 73.3 mmol/g_cat._ after 48 h of reaction.

To study the role of the addition of a cocatalyst to enhance TMM hydration, many kinds of acidic or basic solids were tested, as shown in Figure 5. In the blank test conducted without cocatalyst, 27 and 73.3 mmol DMC/g_cat._ was obtained after 6 h and 48 h, respectively. Basic oxides, MgO and Al_2_O_3_, were first tested as cocatalysts. The results show that DMC production decreased after adding these two basic solids in both 6 h and 48 h reactions. Decomposition of DMC is therefore the dominant process over these basic solids [34]. Regarding the easy hydration of TMM over acidic sites, several acid solids were employed as cocatalysts afterwards. DMC production increased slightly when using SiO_2_ as the second catalyst both after 6 h and 48 h of reaction. Furthermore, acid zeolites containing both Brønsted acid sites and Lewis acid sites were tested. They are interesting because of their adjustable acidity by varying the SiO_2_/Al_2_O_3_ ratio [36]. Therefore, three zeolites presenting different SiO_2_/Al_2_O_3_ ratios were tested as second catalysts. It is important to note that the acid strength of the zeolite increases with an increase in the SiO_2_/Al_2_O_3_ ratio [37,38,39,40]. Here, we used ZSM-5 zeolites, such as Z1 (SiO_2_/Al_2_O_3_ = 280), Z2 (SiO_2_/Al_2_O_3_ = 50) and Z3 (SiO_2_/Al_2_O_3_ = 23). The DMC production observed with these zeolites all decreased compared to the blank test. This can be explained by the increased hydrophobicity of zeolites, presenting high SiO_2_/Al_2_O_3_ ratios. As a consequence, type X zeolites with lower SiO_2_/Al_2_O_3_ ratios were also used as second catalysts. The influence on the DMC production for FAU (SiO_2_/Al_2_O_3_ = 3.2) as a second catalyst is negligible. However, the DMC production with powder 13X (SiO_2_/Al_2_O_3_ = 2.45) is considerably improved both after 6 h (48 mmol/g_cat._) and 48 h (95 mmol/g_cat._), compared to the blank test (27 and 73.3 mmol DMC/g_cat._, respectively). Further decreasing the SiO_2_/Al_2_O_3_ ratio (type A zeolites 5A and LAT, SiO_2_/Al_2_O_3_ = 2) was not beneficial to the DMC production which decreased again. Therefore, 13X was found to be the best option to accelerate the rate of TMM hydration and, consequently, the DMC production. It has to be noted here that 13X has a dual role as a molecular sieve it can also trap water during the reaction. To our opinion, this result is the most important of work as it has never been reported that the combination of two solid catalysts could improve the production of DMC.

As presented in Figure 3, the molar ratio of methanol and TMM at 2:1 produced less DMC as compared to the molar ratio of methanol and TMM at 6:1. However, after adding 13X as a second catalyst, the DMC yield on CeO_2_ catalyst with molar ratio of methanol and TMM at 2:1 enhanced compared to that with high molar ratio of methanol and TMM (6:1), as shown in Figure 6. Therefore, the molar ratio of methanol and TMM at 2:1 is more favourable for the reaction.

Figure 7 shows that the DMC production could further be improved to a certain extent by increasing the mass of the 13X cocatalyst. In total, 45 mg of 13X presented the highest catalytic activity after 6 h of reaction (49.4 mmol/g_cat._). Catalytic activity is diminished when further increasing the mass of the second catalyst. It is believed that DMC was decomposed by an excessive number of acidic sites introduced by 13X. The conversion of TMM increased linearly with an increasing mass of cocatalyst 13X.

### 2.4. Comparison of the Performances of Different Cerias

In order to see the influence of the catalyst, a comparison of DMC production on different types of cerias was performed. Hence, in addition to the CeO_2_ already tested, commercial cerias HSA-5 and HSA-20 possessing different surface areas were also tested in the presence of TMM (molar ratio methanol/TMM = 2:1) and in the presence or absence of 13X (30 mg for 30 mg of catalyst) as cocatalyst. The results are shown in Figure 8.

HSA-5 presents the highest DMC production (101.7 mmol DMC/g_cat._) after 48 h of reaction without adding 13X as a cocatalyst. The addition of 13X as the second catalyst results in significantly higher DMC production for all three catalysts, especially for HSA-20, which gives almost the same DMC production (128.9 mmol/g_cat._) as HSA-5 after 48 h of reaction.

These three ceria-based catalysts were characterized to try to relate their physicochemical properties to their catalytic performances. It is well known that the acidity and basicity of catalysts play a crucial role in the formation of DMC; therefore, NH_3_-TPD and CO_2_-TPD were carried out to measure the acidity and basicity of these three catalysts. The desorption peaks with maxima located at <200 °C, 200–400 °C, and >400 °C are attributed to weak, moderate and strong acidic/basic sites, respectively. They are summarized in Table 1. It is clearly shown that the total amounts of NH_3_/CO_2_ desorbed from acid/base sites for HSA-5 are much higher than that of CeO_2_ and HSA-20, which explains the high DMC production when HSA-5 catalyst is used. Unnikrishnan et al. have shown that the catalytic performances of cerias are related to the concentration of moderate acidic/basic sites [38]. As can be seen in Table 1, the moderate amount of NH_3_/CO_2_ desorbed from acid/base sites for HSA-20 is higher than for CeO_2_, which results in a better DMC production on HSA-20.

As shown in Figure 9, the production of DMC on HSA-5 and HSA-20 increased with reaction time and presented almost the same value after 48 h of reaction. It can be noted that the DMC production obtained in the Parr autoclave is much better than the one obtained in the SPR in the same conditions (namely 199.5 mmol/g_cat._ for HAS-5 and 192.1 mmol/g_cat._ for HAS-20 in the Parr autoclave, compared to 128.4 mmol/g_cat._ for HAS-5 and 128.9 mmol/g_cat._ for HAS-20 using the SPR, respectively). The reason for this improvement is most probably linked to the higher efficiency of the agitation in the Parr autoclave equipped with a Rushton turbine. The best DMC production obtained in the Parr reactor at 130 °C after 48 h reaction over HSA-5 catalyst with 13X as a second catalyst and TMM as dehydration agent is 199.5 mmol/g_cat._.

Table 2 shows a comparison of the catalytic performances of our best catalysts (HSA-5) with other Ce-based catalysts used in the literature for the direct synthesis of DMC from CO_2_ and methanol. The production of DMC obtained in this work is significantly higher than the previous reports, indicating the efficiency of the solid cocatalyst approach.

## 3. Materials and Methods

### 3.1. Materials

Ceria supplied by Daiichi Kigenso Kagaku Kogyo Co. Ltd. is further labelled CeO_2_ (surface area is 152 m^2^/g). Cerias supplied by Solvay company (Brussels, Belgium) are, respectively, labelled HSA-5 (surface area is 235 m^2^/g) and HSA-20 (surface area is 149 m^2^/g). SiO_2_, magnesium oxide (MgO), aluminum oxide (Al_2_O_3_, basic), Molecular sieve 5A powder (60–80 mesh, SiO_2_/Al_2_O_3_ (mol.) = 2), abbreviated as 5A, Zeolite BCR-705 bead (SiO_2_/Al_2_O_3_ (mol.) = 2, micropore width is 0.592 nm), abbreviated as LAT, Zeolite BCR-704 bead (SiO_2_/Al_2_O_3_ (mol.) = 3, micropore width is 0.668 nm), abbreviated as FAU, were purchased from Sigma Aldrich (St. Louis, MO, USA). Zeolite CBV-28014 (SiO_2_/Al_2_O_3_ (mol.) = 280), abbreviated as Z1, Zeolite CBV-5524G (SiO_2_/Al_2_O_3_ (mol.) = 50), abbreviated as Z2, and Zeolite CBV-2314 (SiO_2_/Al_2_O_3_ (mol.) = 23), abbreviated as Z3, were supplied by Zeolyst (Kansas City, KS, USA). A molecular sieve 13X powder was purchased from Chromoptic (Villejust, France, SiO_2_/Al_2_O_3_ (mol.) = 2.45, 100–120 mesh), abbreviated as 13X. All materials were used without further treatment.

The solvent used for the reaction was methanol (≥99.8%, Sigma Aldrich). CO_2_ (99.9%) used as a reagent for the reaction was supplied by Air Liquide. 1,1,1-trimethoxymethane (TMM, for synthesis, Sigma Aldrich), 2,2-dimethoxypropane (DMP, 98%, Sigma Aldrich) and dimethoxymethane (DMM, 99%, Sigma Aldrich) were used as liquid dehydrating agents, and, as such, for GC calibration. Dimethyl carbonate (DMC, 99%, Sigma Aldrich) was used for GC calibration.

### 3.2. Characterization

Temperature-programmed desorption of NH_3_ and CO_2_ (NH_3_-TPD and CO_2_-TPD) was performed to quantify the amount and strength of the acid-base sites of the catalysts. A typical test was as follows: 50 mg catalysts were pre-treated in a He flow (30 mL/min) at 130 °C for 2 h in order to remove the physisorbed water. Then, NH_3_ (CO_2_) was absorbed at the surface by pulsed injections at 130 °C until saturation was stated from the MS signal. The TPD profiles were monitored by MS and thermal conductivity detector and recorded from 130 to 650 °C at a heating rate of 10 °C/min.

### 3.3. Catalytic Evaluation

The catalytic activity for the direct synthesis of DMC from CO_2_ and methanol was evaluated on the REALCAT platform using a Screening Pressure Reactor (SPR) system from Unchained Labs equipped with 24 parallel stainless steel batch reactors of 6 mL each [41]. Typically, 30 mg of CeO_2_ catalyst or/and a given amount of cocatalyst were placed into the reactors with methanol and TMM. Before the reaction, the reactors were purged with CO_2_ several times to remove air and then pressurized up to 20 bar with CO_2_ at room temperature. The reaction temperature and pressure of the reactor were raised to 130 °C and 38 bar for carrying out the reaction for a given time with a continuous stirring of 700 rpm and then cooled down. A scaled-up process was also investigated using a 250 mL Parr autoclave equipped with a Rushton turbine. The reaction conditions were the same as the ones used on the SPR. Reaction products were analyzed using Shimadzu GC-2010 Plus and GC-FID-2010 Plus AF Ultra EI gas chromatographs, equipped with a FID detector, and a ZB-WAX Plus column (Shimadzu, Kyoto, Japan, 30 m × 0.25 mm × 0.25 μm). Solutions after the reaction were filtered to eliminate the solid and analyzed without further dilution.

Since methanol is both a reactant and a product of the hydration of the dehydration agent TMM, calculating the conversion of methanol is impossible. However, TMM conversion and DMC production can be calculated using the following equations:$$TMM\;conversion\;(\%)=\frac{TMM\;consumed\;(mmol)}{TMM\;fed\;(mmol)}\times 100\%$$
$$DMC\;production\;(\frac{mmol}{g.catalyst})=\frac{DMC\;formed\;(mmol)}{Catalyst\;used\;(g)}$$

Note that the mass of the catalyst used in the calculation of the DMC production is the mass of the ceria only and does not include the mass of the cocatalyst.

## 4. Conclusions

In this paper, commercial cerias were investigated as catalysts for the direct conversion of CO_2_ to DMC using methanol as a reactant. Several dehydrating agents were tested in the combination of ceria to shift the thermodynamic equilibrium and hence improve the DMC production. Among them, TMM was found to be the most promising one leading to the highest DMC production (73.3 mmol/g_cat._ after 48 h of reaction). Interestingly, when TMM is hydrolyzed, it yields methanol which is used as a supplementary reactant. However, the hydration reaction was found to be slow in the conditions of the reaction used. Therefore, a new concept was proposed. It consisted of adding a second solid catalyst in the reaction medium to accelerate the TMM reaction with water molecules formed concomitantly to DMC and, hence, shift the equilibrium toward DMC formation. A series of acid and basic cocatalysts were tested and the best option found was a 13X molecular sieve as it provides acidic sites that can accelerate the hydration of TMM. Combining the use of TMM and of 13X molecular sieve as a cocatalyst to the ceria-based catalyst led to a significant increase in the DMC production (49.4 mmol/g_cat._ after 6 h of reaction time), compared to pure ceria (1.1 mmol/g_cat._ after 6 h of reaction time). Different commercial cerias were tested in the same conditions. HSA-5 and HSA-20 presented higher DMC production than CeO_2_. This can be attributed to their high acidic/basic site concentrations. Finally, the reaction process was scaled up in a 250 mL Parr autoclave using HSA-5 and HSA-20 catalysts. Both catalysts presented almost the same DMC production when adding 13X as the second catalyst and when using TMM as a dehydrating agent. The highest DMC production obtained in this work was 199.5 mmol/g_cat._ on HSA-5 catalyst at 130 °C, 38 bar of CO_2_ pressure after 48 h reaction. To the best of our knowledge, this is the highest DMC production published so far in the literature.

## Figures and Tables

**Figure 1 molecules-29-05663-f001:**
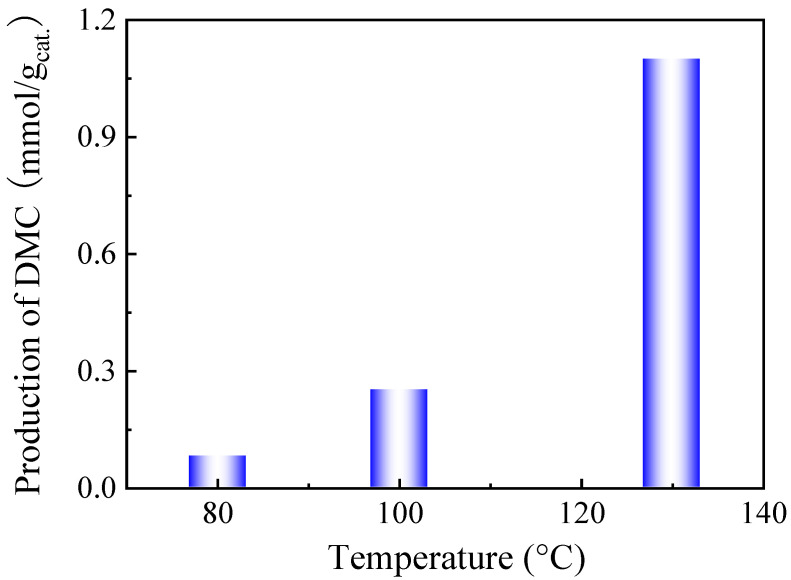
DMC production as a function of the reaction temperature on the CeO_2_ catalyst. Reaction conditions: 2 mL of methanol, 30 mg of CeO_2_, 38 bar CO_2_ pressure, and 6 h reaction time.

**Figure 2 molecules-29-05663-f002:**
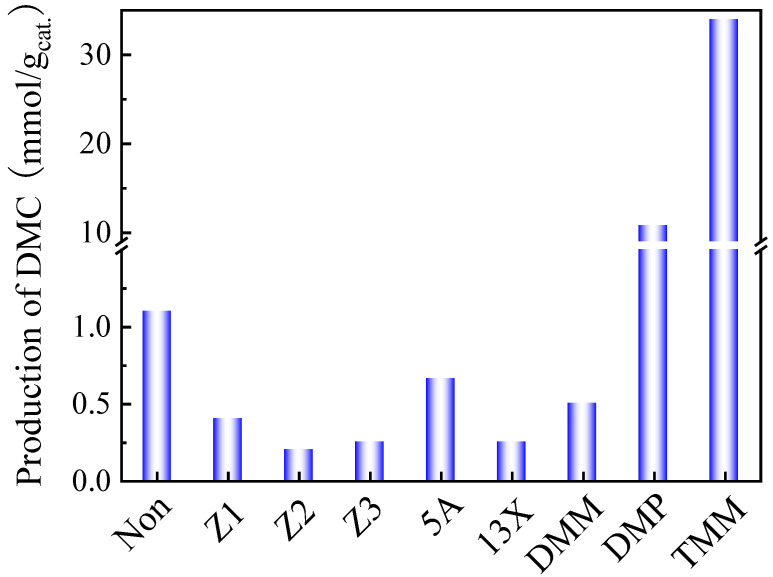
Effect of the addition of different dehydration agents on the production of DMC obtained with CeO_2_ catalyst. Reaction conditions: 2 mL of methanol for solid dehydrants/1 mL of methanol for liquid dehydrants, 30 mg of CeO_2_, 38 bar CO_2_ pressure, 130 °C, reaction time 6 h, 150 mg of solid dehydrants, DMM 0.4 mL (molar ratio of methanol and DMM was 6:1), DMP 0.5 mL (molar ratio of methanol and DMP was 6:1), TMM 0.4 mL (molar ratio of methanol and TMM was 6:1).

**Figure 3 molecules-29-05663-f003:**
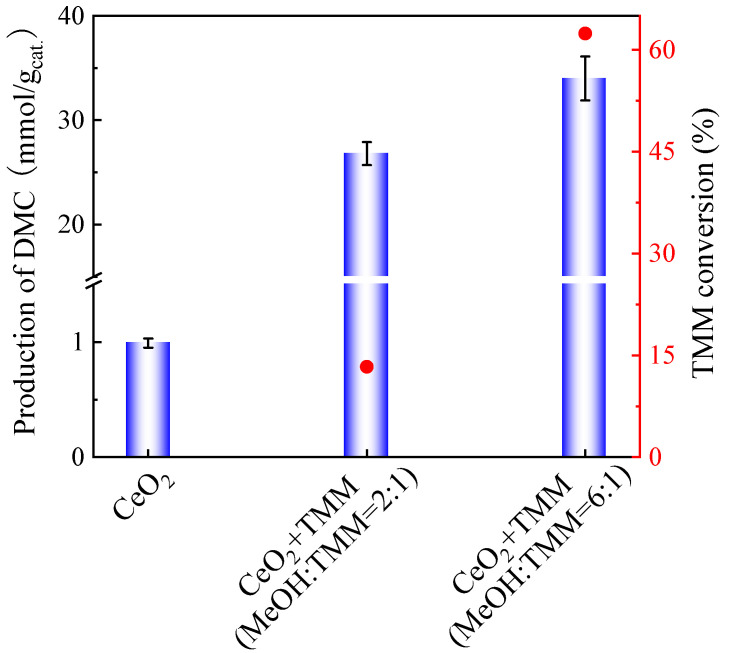
Effect of TMM addition on the production of DMC on CeO_2_ (blue bars) and the TMM conversion (red dots). Reaction conditions: 1 mL of methanol, 0.4 mL (molar ratio of methanol and TMM is 6:1) or 1.3 mL of TMM (molar ratio of methanol and TMM is 2:1), 30 mg of CeO_2_, 38 bar CO_2_ pressure, 130 °C, 6 h reaction time. The error bars correspond to the standard deviation from three independent measurements.

**Figure 4 molecules-29-05663-f004:**
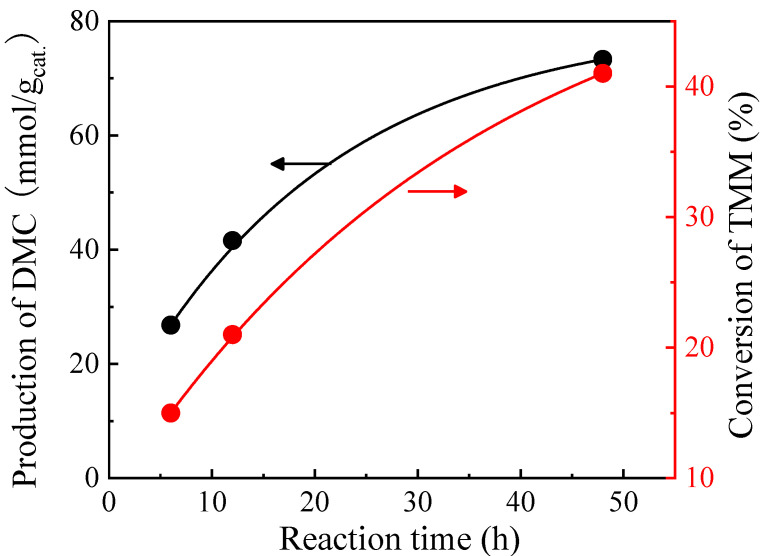
Effect of reaction time on the production of DMC on CeO_2_ catalyst (black curve) and on the conversion of TMM (red curve). Reaction conditions: 1 mL of methanol, 1.3 mL of TMM (molar ratio methanol/TMM = 2:1), 30 mg of CeO_2_, 38 bar CO_2_ pressure, 130 °C.

**Figure 5 molecules-29-05663-f005:**
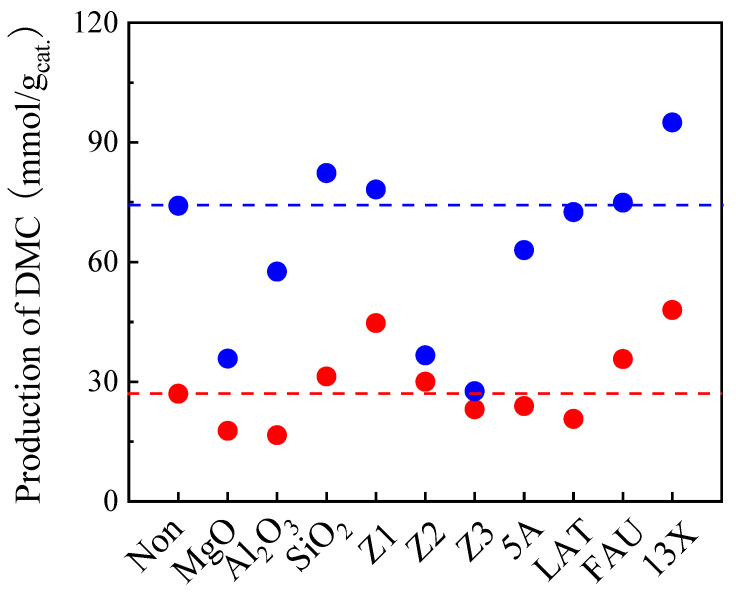
Effect of adding a second catalysts on the DMC production on CeO_2_ with TMM as dehydration agent. Reaction conditions: 1 mL of methanol, 1.3 mL of TMM (molar ratio methanol/TMM = 2:1), 30 mg of CeO_2_, 30 mg of each second catalyst, 38 bar CO_2_ pressure, 130 °C, 6 h (red dots) and 48 h (blue dots). Note: The red and blue horizontal dashed lines represent the DMC production on CeO_2_ with TMM as dehydration agent observed in a blank test done without adding any second catalyst after 6 h and 48 h, respectively.

**Figure 6 molecules-29-05663-f006:**
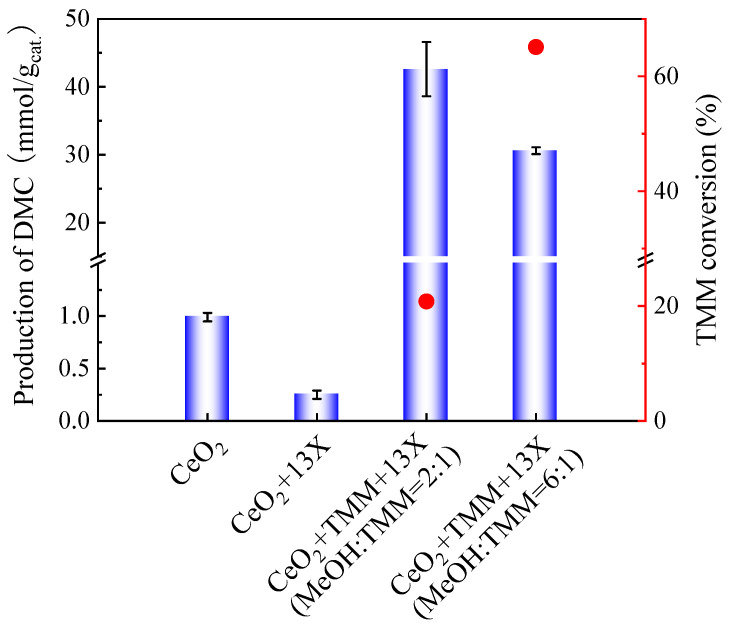
Effect of 13X addition on the production of DMC on CeO_2_ (blue bars) and on the TMM conversion (red dots). Reaction conditions: 1 mL of methanol, 0.4 mL (molar ratio of methanol and TMM is 6:1) or 1.3 mL of TMM (molar ratio of methanol and TMM is 2:1), 30 mg of CeO_2_, 30 mg of 13X, 38 bar CO_2_ pressure, 130 °C, 6 h reaction time. The error bars correspond to the standard deviation from three independent measurements.

**Figure 7 molecules-29-05663-f007:**
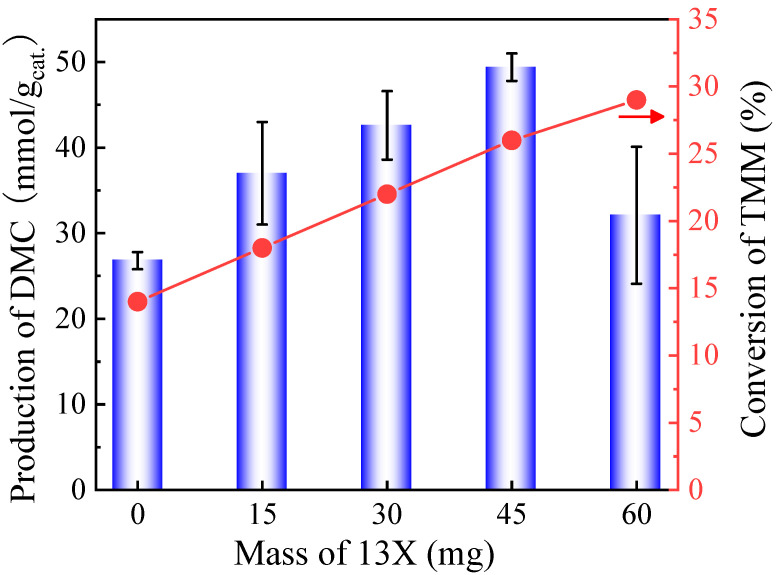
DMC production and TMM conversion as a function of the mass of 13X added on CeO_2_ catalyst using TMM as dehydration agent. Reaction conditions: 1 mL of methanol, 1.3 mL of TMM (molar ratio methanol/TMM = 2:1), 30 mg of CeO_2_, 38 bar CO_2_ pressure, 130 °C, 6 h reaction time. The error bars correspond to the standard deviation from three independent measurements.

**Figure 8 molecules-29-05663-f008:**
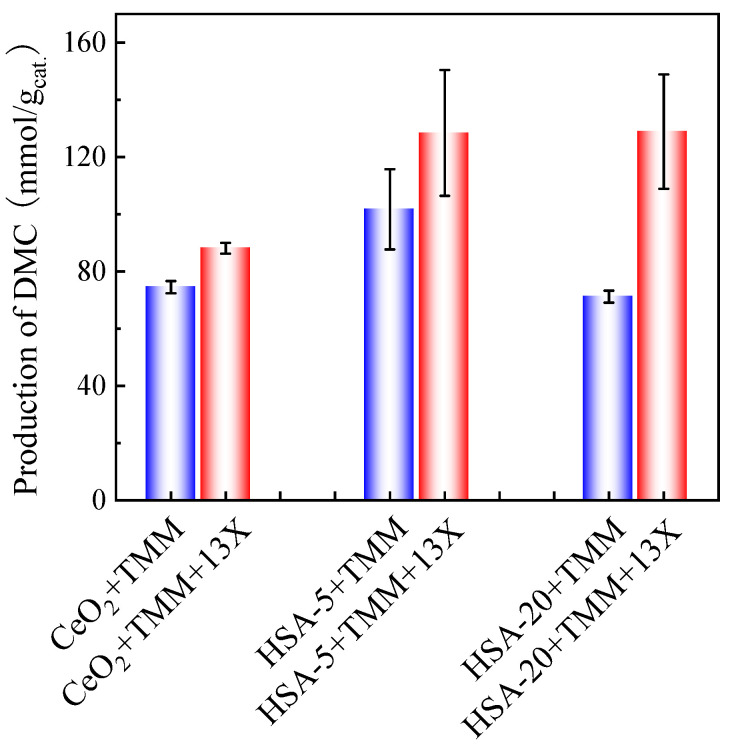
Performances of different ceria-based catalysts using TMM as a dehydration agent and in the presence (red bars) or absence (blue bars) of 13X as cocatalyst. Reaction conditions: 1 mL of methanol, 1.3 mL of TMM (molar ratio methanol/TMM = 2:1), 30 mg of catalyst, 30 mg of 13X (when present), 38 bar CO_2_ pressure, 130 °C, 48 h reaction times. The error bars correspond to the standard deviation from three independent measurements.

**Figure 9 molecules-29-05663-f009:**
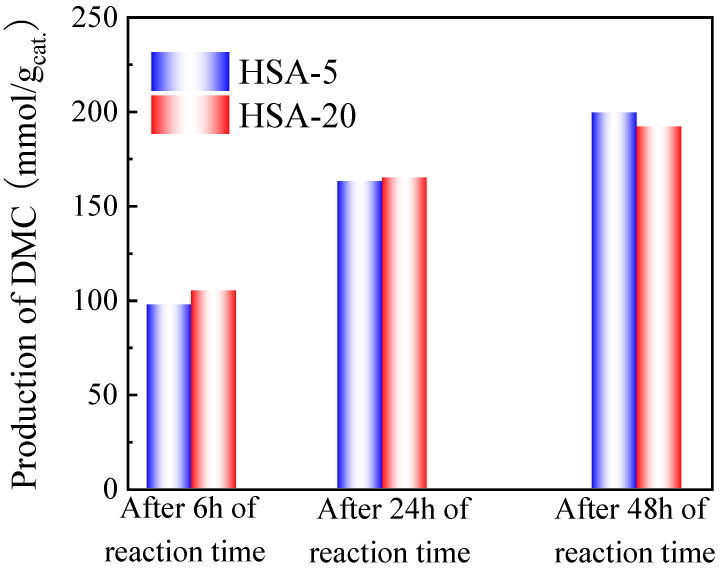
Effect of reaction time on the production of DMC on HSA-5 and HSA-20 catalysts with TMM as dehydration agent performed on a 250 mL Parr autoclave. Reaction conditions: 60 mL of methanol, 78 mL of TMM (molar ratio methanol/TMM = 2:1), 1.8 g of catalyst, 1.8 g of 13X, 130 °C, 38 bar CO_2_ pressure, 6 h, 24 h and 48 h reaction times.

**Table 1 molecules-29-05663-t001:** Acidic and basic amounts of cerias determined from CO_2_-TPD and NH_3_-TPD measurements.

Samples	NH_3_ Absorption (μmol/m^2^) ^a^				CO_2_ Absorption (μmol/m^2^) ^b^			
	Weak (<200 °C)	Moderate (200–400 °C)	Strong (>400 °C)	Total	Weak (<200 °C)	Moderate (200–400 °C)	Strong (>400 °C)	Total
CeO_2_	0.2	0.8	2.9	3.9	1.0	1.5	1.1	3.6
HSA-5	0	5.3	7.0	12.3	0.2	1.9	3.0	5.1
HSA-20	0.2	1.6	2.3	4.1	0.8	1.6	0.9	3.3

^a^ Determined from NH_3_-TPD measurement. ^b^ Determined from CO_2_-TPD measurement.

**Table 2 molecules-29-05663-t002:** Comparison of our best results with the catalytic performances obtained over ceria-based catalysts using organic dehydrating agents as reported in the literature.

Entry	Catalyst	Dehydrating Agent	Temp. (°C)	Time (h)	DMC Production (mmol/g_cat._)	Ref.
1	CeO_2_-ZrO_2_	DMP	110	140	13.8	[29]
2	CeO_2_	Benzonitrile	150	24	48.2	[28]
3	CeO_2_	Acetonitrile	150	24	24	[27]
4	Ce-Zr oxide/graphene	TMM	110	24	51.6	[30]
5	CeO_2_	2-cyanopyridine	120	12	138	[21]
6	HSA-5	TMM	130	48	199.5	This work

## Data Availability

Data are contained within the article.

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
