# Peer review of "Ceria-Catalysed Production of Dimethyl Carbonate from Methanol and CO2: Effect of Using a Dehydrating Agent Combined with a Solid Cocatalyst"

_molecules, 2024, doi:10.3390/molecules29235663_

Round 1
Reviewer 1 Report
Comments and Suggestions for Authors
The study explores the use of a second solid cocatalyst (13X molecular sieve) to enhance TMM hydration without degrading DMC. While the research adheres to conventional methodologies within catalysis research, it seems lack novel insights that significantly advance the field. The presentation and organization of the work could be improved for clearer understanding, and the analysis of results requires a deeper, more rigorous scientific examination. To strengthen the paper, a more detailed discussion of results and their implications is necessary. After addressing these points, the manuscript might be suitable for publication in Molecules. Below are specific comments and recommendations for enhancement.
1. Minor typographical, formatting, and grammatical issues require attention to enhance clarity and precision.
2. What strategies can be proposed to overcome these challenges, and how can the novelty of the approach be emphasized to establish a stronger scientific foundation for the work?
3. Could you please advise on how to enhance the cohesion of the introduction and provide a more concise explanation of the study's rationale, specifically by detailing the existing challenges in the field?
4. How does COâ‚‚ pressure affect the production of DMC at high temperatures?
5. What specific properties of TMM make it a more effective dehydrating agent than solid options like Z1, Z2, Z3, 5A, and 13X in promoting DMC production, and how does its hydrolysis contribute to this effect?
6. What are the possible mechanisms of DMC decomposition in the presence of highly acidic solid dehydrating agents?
7. How does the use of a secondary catalyst influence the overall reaction mechanism, and what considerations are necessary to ensure compatibility?
Author Response
Please see the attached file with our detailed answers.

Reviewer 2 Report
Comments and Suggestions for Authors
The article is devoted to an important topic dedicated to the synthesis of DMC from available starting compounds - methanol and CO2. The article is well written and systematized. The authors consistently optimized the reaction conditions for obtaining dimethyl carbonate by testing conditions from the process temperature to the type of commercial cerium oxide and the second catalyst. Quite promising conditions with the highest DMC production in the literature were found. The article may be accepted after minor revision.
Additional points:
1) Please add methanol conversion (composition of final mixtures) and reusability experiments of the best catalytic system.
2) It is easily to obtain DMC in pure form from these mixtures? Please suggest it by some experimental data or references.
3) More accurate writing of conclusion concerning achieved results is needed.
Author Response

(The authors gave the same response as above.)

Round 2
Reviewer 1 Report
Comments and Suggestions for Authors
All my questions are settled and the revised manuscript is fine for publication.
Reviewer 2 Report
Comments and Suggestions for Authors
The revision is satisfactory, I suggest to accept the manuscript in its current form.